# Engineering Skeletal Muscle Grafts with PAX7::GFP-Sorted Human Pluripotent Stem Cell-Derived Myogenic Progenitors on Fibrin Microfiber Bundles for Tissue Regeneration

**DOI:** 10.3390/bioengineering9110693

**Published:** 2022-11-15

**Authors:** Sarah M. Somers, Jordana Gilbert-Honick, In Young Choi, Emily K. W. Lo, HoTae Lim, Shaquielle Dias, Kathryn R. Wagner, Hai-Quan Mao, Patrick Cahan, Gabsang Lee, Warren L. Grayson

**Affiliations:** 1Translational Tissue Engineering Center, Johns Hopkins University School of Medicine, Baltimore, MD 21231, USA; 2Department of Biomedical Engineering, Johns Hopkins University School of Medicine, Baltimore, MD 21205, USA; 3The Institute for Cell Engineering, Johns Hopkins University School of Medicine, Baltimore, MD 21205, USA; 4Department of Pathology, Graduate School, Kyung Hee University, Seoul 02447, Republic of Korea; 5College of Veterinary Medicine, Chungbuk National University, Chungbuk 28644, Republic of Korea; 6The Hugo W. Moser Research Institute, Kennedy Krieger Institute, Baltimore, MD 21205, USA; 7Graduate Program in Cellular and Molecular Medicine, Johns Hopkins University School of Medicine, Baltimore, MD 21205, USA; 8Department of Neurology, Johns Hopkins University School of Medicine, Baltimore, MD 21205, USA; 9The Solomon H. Synder Department of Neuroscience, Johns Hopkins University School of Medicine, Baltimore, MD 21205, USA; 10Institute for NanoBioTechnology (INBT), Johns Hopkins University School of Engineering, Baltimore, MD 21218, USA; 11Department of Material Sciences & Engineering, Johns Hopkins University School of Engineering, Baltimore, MD 21218, USA; 12Department of Chemical & Biomolecular, Johns Hopkins University School of Engineering, Baltimore, MD 21218, USA

**Keywords:** human pluripotent stem cells (hPSCs), hPSC-derived myogenic progenitors (hPDMs) PAX7, skeletal muscle tissue engineering, skeletal muscle injury and regeneration, fibrin microfiber bundles

## Abstract

Tissue engineering strategies that combine human pluripotent stem cell-derived myogenic progenitors (hPDMs) with advanced biomaterials provide promising tools for engineering 3D skeletal muscle grafts to model tissue development in vitro and promote muscle regeneration in vivo. We recently demonstrated (i) the potential for obtaining large numbers of hPDMs using a combination of two small molecules without the overexpression of transgenes and (ii) the application of electrospun fibrin microfiber bundles for functional skeletal muscle restoration following volumetric muscle loss. In this study, we aimed to demonstrate that the biophysical cues provided by the fibrin microfiber bundles induce hPDMs to form engineered human skeletal muscle grafts containing multinucleated myotubes that express desmin and myosin heavy chains and that these grafts could promote regeneration following skeletal muscle injuries. We tested a genetic PAX7 reporter line (PAX7::GFP) to sort for more homogenous populations of hPDMs. RNA sequencing and gene set enrichment analyses confirmed that PAX7::GFP-sorted hPDMs exhibited high expression of myogenic genes. We tested engineered human skeletal muscle grafts derived from PAX7::GFP-sorted hPDMs within in vivo skeletal muscle defects by assessing myogenesis, engraftment and immunogenicity using immunohistochemical staining. The PAX7::GFP-sorted groups had moderately high vascular infiltration and more implanted cell association with embryonic myosin heavy chain (eMHC) regions, suggesting they induced pro-regenerative microenvironments. These findings demonstrated the promise for the use of PAX7::GFP-sorted hPDMs on fibrin microfiber bundles and provided some insights for improving the cell–biomaterial system to stimulate more robust in vivo skeletal muscle regeneration.

## 1. Introduction

Skeletal muscle is known to have a high regenerative capacity. However, this capacity can be overwhelmed when large volumes of tissue are removed as is the case in volumetric muscle loss injuries [1,2]. Tissue engineering is an attractive approach for the regeneration of skeletal muscle tissues for studying critical elements of tissue development in vitro, as well as for treating clinical indications, such as volumetric muscle loss (VML) [1,2], i.e., injuries characterized by losses of greater than 20% of a muscle’s volume [1]. VML injuries are associated with significant morbidity without corrective muscle flaps [2]. However, the flaps provide little functional improvement, and their use is limited by the donor site morbidity [1,2]. The most effective tissue engineering strategies include biomaterial scaffolds for directing the 3D structure of the de novo tissue and a pro-regenerative cell source to assist in the formation of functional tissue. Both synthetic [3,4,5,6,7,8] and naturally occurring [3,5,8,9,10,11,12,13] materials have been tested for the treatment of VML and tuned to mimic the mechanics and structure of skeletal muscle, as well as to exhibit biodegradability, biocompatibility, bioactivity, and minimal batch-to-batch variability and immunogenicity. We have previously developed a 3D scaffold composed of electrospun fibrin microfiber bundles [13], which mimic the native mechanical stiffness of skeletal muscle; induce cell alignment, allowing for the formation of aligned, striated myotubes from C2C12 mouse myoblasts; and are sufficiently robust to be sutured in place [3,14,15]. In this work, we tested whether we could engineer 3D skeletal muscle grafts with human stem cell-derived myoprogenitors and the fibrin microfiber bundles to promote histological regeneration of skeletal muscle tissue.

Several human cell types have been tested for the formation of a tissue engineered skeletal muscle graft, each with their own challenges: myoblasts and satellite cells [16,17,18,19] have good differentiation capabilities but are difficult to isolate and expand in culture to numbers that would be clinically relevant for treatment [11,20]. Adult stem cell populations, such as mesenchymal stem cells [6,21] and adipose-derived stem cells [7,8,22], have key clinical advantages, including the ability to be patient-specific and expandable in culture, but their low differentiation efficiency hinders the functional regeneration of skeletal muscle tissue [20,22]. Human pluripotent stem cells (hPSCs) provide an attractive cell source for myoblasts due to their capability to differentiate into primary myoblasts and some supporting cells along the myogenic lineage [4,5]. Therefore, hPSCs have been used for skeletal muscle grafts [4,23] as they can be patient-specific and highly expandable, can differentiate down the myogenic lineage [10,11], and have been shown to be a highly promising cell source for skeletal muscle regeneration. However, previous studies have demonstrated the ability to differentiate hPSCs down the myogenic lineage for 3D constructs implanted in vivo, utilizing the transgenic overexpression of myogenic genes, MyoD [23] and PAX7 [4], resulting in improved differentiation and repeatability of the technique.

Although promising for in vitro assays, the genetic modification of hPSCs has limited translatability due to its ability to cause genetic abnormalities [10,11]. In previous work, the ability to commit hPSCs towards human pluripotent stem cell-derived myogenic progenitors (hPDMs) was demonstrated using a combination of two small molecules, CHIR99021 and DAPT, creating an expandable cell source capable of forming myotubes in 2D and engrafting in vivo [12]. More recently, we purified this cell population using a reporter line for PAX7+ cells that allowed for the sorting of this myogenic population [5]. Hence, the goal of this study was to investigate the use of purified PAX7::GFP-sorted hPDMs in 3D in vitro culture and in in vivo muscle defects to determine their myogenic regeneration capabilities, in combination with electrospun fibrin microfiber bundles. We characterized transgene-free hPDMs seeded onto fibrin microfibers using immunohistochemistry to highlight the transcriptomic efficiency in vitro then characterized engraftment efficiency and immunogenicity in vivo. Implanted 3D constructs were able to engraft with moderate fibrosis during early maturation. This investigation demonstrates that hPDMs grown in fibrin-based 3D culture have a predominantly pro-regenerative immune response with minimal fibrosis during early muscle regeneration.

## 2. Materials and Methods

### 2.1. Electrospinning Fibrin Scaffolds

Fibrin scaffolds were electrospun in a sterile environment, as previously described [3,13]. In brief, the fibrin microfiber bundles were created through a co-extrusion of syringes of 1% fibrinogen (Sigma-Aldrich, St. Louis, MO, USA) in 0.2% polyethylene oxide (PEO, average Mv ~ 4 000 kDa, Sigma-Aldrich) and 0.75% alginate (Sigma-Aldrich) in 0.2% PEO. The solutions were combined through a y-syringe and passed through a 27G needle-tip with 3–5 kV of applied voltage. The resulting stream was collected on a rotating dish (~35 rpm), containing 50 mM CaCl_2_ and 20 U/mL thrombin (Sigma-Aldrich) to crosslink the scaffold for 5.75 min. The scaffold was allowed to crosslink for an additional 3–5 min and was wrapped 4 times around an acrylonitrile-butadiene-styrene (ABS) frame. Following this, scaffolds were incubated overnight in 250 mM sodium citrate (Sigma-Aldrich) to dissolve the alginate, resulting in a 700–1000 µm diameter fibrin microfiber bundle that could be transferred and stored in DI water for up to 2 weeks before use.

### 2.2. The Myogenic Commitment and Expansion of hPSC

The H9 hESC cell line was provided by WiCell Research Institute Inc and used in accordance with an institutional review board-approved protocol. As previously described, human embryonic stem cells (H9 hESCs; WiCell, Madison, WI, USA) were cultured and committed towards the myogenic lineage [5,12]. Briefly, before commitment, hESCs were expanded using a mouse embryonic fibroblast (MEFs; GlobalStem, Rockville, MD, USA or AppliedStem-Cell, Milpitas, CA, USA) feeder layer and expansion media containing DMEM/F12, 20% knockout serum replacement, 1 mM L-glutamine, 100 mM MEM non-essential amino acids, 0.1 mM b-mercaptoethanol, and 10 ng/mL FGF2. For commitment, N2 media, which contained DMEM/F12 powder, glucose, sodium bicarbonate, insulin, putrescine, progesterone, sodium selenite, and transferrin, was conditioned by MEFs for 24 h and sterile filtered. Commitment was performed with PAX7::GFP-hESCs (following genetic manipulation with the CRISPR/Cas9 system to express a PAX7::GFP reporter [5]) seeded in a 24-well culture plate coated with 1% Geltrex at a density of 1.5 × 10^5^ cells/well. For the first 24 h, the hESCs were fed MEF-conditioned N2 media containing 10 ng/mL FGF2 and 10 µM Y-27632 (Cayman Chemical, Ann Arbor, MI, USA). For induction, on days 1–4 the N2 media was supplemented with 3 μM CHIR99021 (Cayman Chemical), and on days 4–12 the N2 media was supplemented with 10 μM DAPT (Cayman Chemical). Following approximately 30 days of commitment, the resulting hPDMs sorted for using fluorescence-activated cell sorting (FACS) and were expanded by 5–6 passages in N2 media supplemented with 5% FBS, at which point the cells were seeded on scaffolds.

### 2.3. Cell Seeding on Fibrin Microfiber Bundle Scaffolds

For optimal cell adhesion to the fibrin microfiber bundles, PAX7::GFP-sorted hPDMs were suspended in thrombin at 1.25 U/mL in N2 media containing 12.5 mmol/L CaCl2. Fibrinogen was diluted in N2 media to 20 mg/mL and equal parts of fibrinogen and the thrombin cell mixture were mixed and immediately pipetted onto the scaffold surface in 5 µL increments. Cells were seeded at a concentration of 300,000 cells/fiber on short fibers (15 mm) or 600,000 cells/fiber on long fibers (30 mm). Following seeding, constructs were incubated for 1 h before the feeding media was applied. Halfway through this incubation, 15 µL of media was added to maintain sample hydration. The hPDMs were cultured in N2 media with 5% FBS, 1% P/S, and 30 µg/mL aprotinin (Affymetrix, Santa Clara, CA, USA). PAX7::GFP-sorted hPDMs were cultured in PAX7 media, which contains N2 media supplemented with 5% FBS, 1% P/S, 10 ng/mL FGF2 (PeproTech, Rocky Hill, NJ, USA), and 100 ng/mL FGF8 (PeproTech). Media was exchanged every 2 days for all studies.

### 2.4. Whole Mount Immunostaining

Following culture, samples were prepared for whole-mount immunostaining by fixing in 4% paraformaldehyde at 4 °C for 3 h on a rocker at 80 rpm. Constructs were then washed with PBS 3 times and placed in block/permeabilization solution containing 0.2% Triton X-100 and 10% normal goat or donkey serum for 3 h. Samples were then incubated in primary antibody solution overnight, containing 5% normal serum and 0.1% Tween 20 in PBS at 4 °C on a rocker at 135 rpm. Primary antibodies used were: myogenin (5 μg/mL; DSHB, Iowa City, IA, USA), myosin heavy chain, MF20 (5 μg/mL; DSHB), desmin (1:400; Abcam, Cambridge, UK), α-actinin (1:200; Sigma-Aldrich), titin (5 μg/mL; DSHB), and PAX7 (5 μg/mL; DSHB). The following day, 3 × 1 h washes were performed in PBS containing 0.1% Tween 20. The samples were then incubated overnight at 4 °C in secondary solution containing 5% normal serum and 0.1% Tween 20 in PBS on a rocker at 135 rpm. The secondary antibodies used included: DyLight 488-conjugated goat anti-mouse, DyLight 649-conjugated goat anti-rabbit, Alexa Fluor 647-conjugated donkey anti-rabbit, or Cy3-conjugated donkey anti-mouse (1:400; Jackson ImmunoResearch, West Grove, PA, USA). This incubation was again followed by 3 × 1 h washes in PBS containing 0.1% Tween 20. DAPI was included with the secondaries or in one of the following washes. Samples were then imaged on a Zeiss LSM 510 confocal microscope, except for the full fiber image, which was imaged on a Zeiss Axio Observer 7 with apotome capabilities.

### 2.5. Animal Models

All animal and surgical procedures were performed in accordance with the Institutional Animal Care and Use Committee at Johns Hopkins University School of Medicine. For all animal experiments female NOD-scid IL2Rgnull (NSG) immunodeficient mice (Jackson Lab, Bar Harbor, ME, USA) aged 2–4 months (*n* = 9) were used to allow for implantation of human cells. Sub-critical “pocket” defects (~5–13% of TA muscle) were performed, as previously described [3]. Briefly, following isoflurane anesthetization, the TA was exposed, and part of the muscle was removed. Engineered muscle grafts were then placed in the defect site (2 skeletal muscle grafts per defect) and sutured to the remaining muscle using nonabsorbable sutures (6–0 Nylon, Express Medical Supplies). Implanted scaffolds were grown for 9–12 days prior to implantation. Sutures, as well as surgical glue (Histoacryl, B Braun Medical, Bethlehem, PA, USA), were also used to close the skin and Rimadyl (Patterson Veterinary, Greeley, CO, USA) (5 mg/kg) was injected subcutaneously after the surgery for pain management. The mice were sacrificed by isoflurane overdose and cervical dislocation 1 to 2 weeks after the surgery. Immediately following this, the TA muscle, including the implanted scaffolds, was removed and cryopreserved for sectioning.

### 2.6. Histology

Following cryopreservation of the TA muscle, samples were sectioned on a cryostat (Leica) in either cross-section (10 μm thickness) or longitudinal section (70 μm). Trichrome staining was done following the manufacturer’s instructions. Immunohistochemistry was performed by fixing sectioned slides in cold 4% paraformaldehyde or ice-cold acetone followed by 3x PBS washes. Blocking and permeabilization was performed by incubating slides in 10% normal goat or donkey serum and 0.2% Triton X-100 at RT for 1 h. Bovine serum albumin (BSA) was also included in blocking for eMHC/LamAC/CD31 staining, as well as for macrophage staining. For samples stained with F4/80, additional streptavidin and biotin blocking was performed. Samples were then incubated in primary antibody solution containing 5% normal serum and PBS overnight at 4 °C. Primary antibodies were used as follows: mouse anti-myosin heavy chain, MF20 (5 μg/mL; DSHB), desmin (1:400; Abcam), rat anti CD31 (1:50; BD Biosciences, San Jose, CA, USA), goat anti-mouse CD31 (1:100; R&D Systems), rabbit anti-LaminAC (1:500; Abcam), mouse anti-PAX7 (5 μg/mL; DSHB), mouse anti embryonic myosin (5 μg/mL; DSHB), rabbit anti-CD86 (1:50; Thermo Fisher Scientific, Waltham, MA, USA), biotin rat anti-mouse F4/80 (1:50; BioLegend), and goat anti-CD206 (1:50, Santa Cruz Biotechnology, Dallas, TX, USA). Slides were then washed 3 times in PBS and incubated for an hour at RT in secondary antibody dilutant containing 5% normal serum. The secondary antibodies used were as follows: Cy3-conjugated donkey anti-mouse, Alexa Fluor 488-conjugated donkey anti-goat, DyLight 488-conjugated goat anti-mouse, DyLight 649-conjugated goat anti-rabbit, AF594-conjugated goat anti-rat, or Alexa Fluor 647-conjugated donkey anti-rabbit (1:400; Jackson ImmunoResearch), as well as Streptavidin AF647 conjugate, donkey anti-rabbit AF555 plus, donkey anti-rabbit AF647 plus, donkey anti-goat AF488 plus, donkey anti-rabbit AF488 plus, donkey anti-goat AF555 plus, donkey anti-mouse AF555 plus (1:300 to 1:600; Thermo Fisher). Alexa Fluor 488 Phalloidin (1:40; Thermo Fisher), when used, was incorporated into secondary stains, as was DAPI (1:2000). Slides were then washed 3 times with PBS and mounted in 50% glycerol/50% PBS for imaging on the Zeiss Axio Observer 7 (Zeiss, Oberkochen, Germany) with apotome capabilities or the Zeiss LSM 510 confocal microscope (Zeiss, Oberkochen, Germany).

### 2.7. Image Quantification

All image quantification was performed using custom ImageJ (NIH, Rockville, MD, USA) macros or MATLAB scripts. MHC area coverage was calculated by first thresholding each image and then measuring the percentage area covered by positive stain. This positive region was then overlayed with the thresholded DAPI stain and the percentage of MF20-associated nuclei was a measure of the percent of the positive DAPI pixels that fell within the MF20 thresholded region. Quantification of PAX7+ cells was performed using a custom MATLAB script that defined all DAPI regions and counted cells as PAX7+ if PAX7 staining was present in the region. Desmin and MF20 coverage were measured from thresholded images. PAX7+ cells, desmin coverage, and MF20 coverage were quantified from PAX7::GFP-sorted hPDMs grown on Geltrex-coated fibers. ImageJ was used to quantify the fibrosis within the defect region by measuring the collagen-dense positive area (*n* = 8, 4 samples with 2 cryosections per sample). Masson’s trichrome images were deconvoluted using ImageJ software, the cross-sectional area of collagen-dense regions quantified then normalized to defect cross-sectional area.

### 2.8. Bulk RNA Sequencing

RNA from 3 separate batches of passaged unsorted hPDMs and 3 batches of passaged PAX7::GFP-sorted hPDMs were isolated manually using TRIzol. RNA library preparation and sequencing were performed by the Johns Hopkins Transcriptomics and Deep Sequencing Core Facility. Libraries were prepared with 200 ng of RNA from each sample with the Illumina^®^ Stranded Total RNA Prep Ligation with Ribo-Zero Plus Kit, following manufacturer’s instructions. Sequencing was performed on the Illumina NovaSeq 6000 with 100 cycles and 800 million reads. Raw expression data are available on GEO under accession [GSE178784].

### 2.9. RNA Sequencing Analysis

Reads were trimmed using cutadapt [24]. Trimmed reads were mapped and quantified against human reference genome and transcriptome-using salmon [25]. Salmon index was generated using human genome assembly GRCh37.p13 and GENCODE human transcriptome assembly Release 19. For quality control, 1e5 reads were randomly sampled from each trimmed FASTQ file and aligned to human genome assembly GRCh37 using HISAT2 [26]. Read feature mapping was performed using htseq-count [27] and GEN-CODE Release 19 comprehensive gene annotation GTF.

Differential gene expression analysis was performed on salmon-generated gene counts using Bioconductor package DESeq2 [28] version 1.30.0. Genes with an FDR-corrected *p*-value < 0.01 were considered significantly differentially expressed between sorted and unsorted hPDMs. To assess for myogenic differentiation and hallmark markers of myogenesis, pre-ranked gene set enrichment analysis (GSEA) was performed using Bioconductor package FGSEA [29] version 1.16.0 with hallmark and C5 gene ontology (GO) gene sets from the Molecular Signatures Database (MSigDB) [30,31]. DESeq2 ‘stat’ was used to pre-rank genes, and fgseaSimple was run with the following parameters: minSize = 3, maxSize = 500 (recommended maximum to cap run time), nperm = 50,000 (to allow a minimal nominal *p*-value of approximately 1/nperm, or in this case 2 × 10^−5^).

To assess hPDM cell type identity, we used singleCellNet [32] to train a cross-species classifier on 14 tissues selected from a mouse organogenesis cell atlas based on single-cell RNA-seq of 61 embryos between days E9.5 to E13.5 [33]. The classifier was trained with the following parameters: training ncells = 3000, nTopGenes = 25, nRand = 200, nTrees = 1000, nTopGenePairs = 50, validation ncells = 500.

### 2.10. Statistics

Statistical analyses were performed using GraphPad Prism 5 software. In cases of direct comparisons, t-tests were used to determine significance while one-way and two-way ANOVAs were used for data with multiple comparisons followed by a Tukey or Bonferroni post-test. Error bars represent standard deviation (SD) unless otherwise noted. *: *p* < 0.05; **: *p* < 0.01; ***: *p* < 0.001.

## 3. Results

### 3.1. Myogenic Induction, Expansion, and Engraftment of Human Pluripotent Stem Cells

We established a small molecule, transgene-free commitment protocol (Figure 1A) for the creation of hPDMs. These progenitors were culture expanded for up to passage 5 or 6 but inevitably resulted in a heterogenous population, which was Pax7-sorted for myogenic progenitors. Electrospun fibrin microfiber bundles (Figure 1B) provided a biomimetic framework for myoblast expansion and myotube formation. These microfiber bundles were previously demonstrated to have stiffness comparable to skeletal muscle and adequate porosity for cell infiltration and proliferation. Pax7-sorted hPDMs were culture expanded for several passages, encapsulated within a bulk hydrogel, pipetted onto the microfiber bundles (Figure 1C), and grown for 10 days before analysis.

### 3.2. Pax7-Sorted Myogenic Cells Form 3D Skeletal Muscle Constructs

To engineer muscle grafts, we grew PAX7::GFP-sorted hPDMs on fibrin microfiber bundles for 10 days. Early in the growth period (day 3), 47.1 ± 14.4% of the PAX7::GFP-sorted hPDMs expressed PAX7 (Figure 2A). By day 10, the PAX7::GFP-sorted hPDMs fused to form mature myotubes expressing MF20 and desmin (Figure 2B). The expression of these markers of mature skeletal muscle was consistent along the length of the fiber (Figure 2C). Striated myotubes expressing α-actinin and titin were also present on the fibers (Figure 2D,E). The growth of the PAX7::GFP-sorted hPDMs on the fibers mimicked the maturation profile of native muscle. The expression of PAX7 decreased significantly over the 10-day growth period, suggesting that the number of myogenic precursors decreased with extended culture. Simultaneously, desmin and MHC coverage significantly increased over the growth period (Figure 2F), indicating that the cells were exhibiting more mature myogenic phenotypes on the fibrin microfiber bundles even in the absence of differentiation medium. Together, the results point to alignment and myogenic maturation of hPDMs in vitro, since the depletion of the myogenic progenitors correlates with increases in the presence of mature myotubes. Additionally, the lag in MHC positivity relative to desmin coverage points to cytoskeletal maturation (including alignment) to support myogenic maturation.

### 3.3. The Expression Profile of PAX7::GFP-Sorted hPDMs was Enriched in Myogenic Programs

Following commitment, the hPDMs were sorted for the GFP+ population and passaged several times to acquire a population of cells sufficient for seeding 3D fibrin microfiber bundles. We first determined the transcriptional differences in unsorted hPDMs and PAX7::GFP-sorted hPDMs by performing bulk RNA sequencing. Principal component analysis indicated that 98.4% of variation was explained by the first principal component, which separated samples by condition (PAX7::GFP-sorted vs. unsorted) (Figure 3A), with *KRT80*, *S100A6*, *F13A1*, and *SERPINB7* as the top positive loadings for PC1 and the *RTL1*, *COL2A1*, *CRABP1*, and *CHGA* as the top negative loadings. Given this clear distinction between the two populations, we globally assessed the identity of the cell populations present in the unsorted and PAX7::GFP-sorted hPDMs using singleCellNet—a computational tool to compare the expression profiles of engineered cells to those of native in vivo populations [32]. We trained a cross-species singleCellNet classifier on 3000 cells each for 14 cell types selected from a fetal mouse organogenesis atlas (days E9.5–E13.5) [33] and validated the classifier on 500 held-out cells per cell type (Appendix A). When queried against the classifier, both PAX7::GFP-sorted hPDMs and unsorted hPDMs classified mainly as myocytes (with background signatures in the cardiac muscle lineage). PAX7::GFP-sorted hPDMs achieved a higher average myocyte classification score than the unsorted hPDMs (Figure 3B).

To further examine the transcriptional differences between PAX7::GFP-sorted and unsorted hPDMs, we performed differential expression analysis, which identified 13,755 significantly differentially expressed genes; notably, *ANO2*, *AHNAK2*, *ANKRD1*, and *S100A16* are significantly more highly expressed in the sorted population and associated with Ca^2+^ signaling or proteins that interact with sarcomeric proteins (Figure 3C). We also examined the expression of marker genes from lineages of interest more specifically, including embryonic, myogenic, neural crest, and fibro–adipogenic progenitor (FAP) cells. We observed increased expression of embryonic, neural crest, and FAP-associated genes in the unsorted hPDMs, while several myogenic and Ca^2+^ handling genes were upregulated in the PAX7::GFP-sorted hPDMs (Figure 3D and Appendix A). To elucidate broader transcriptional trends among the identified differentially expressed genes, we performed gene set enrichment analysis (GSEA) using the C5 gene ontology (GO) gene sets and the hallmark gene sets from the Molecular Signatures Database (MSigDB) (Appendix A). The Hallmark Myogenesis gene set (including *TNNI1, ACTN2, TNNC1, MYO1C, TNNT1, MYLPF, MEF2C,* and *MYH2*) and GO Skeletal Muscle Cell Differentiation gene set (including AKIRIN1 and MSTN) were enriched in the PAX7::GFP-sorted hPDMs (Figure 3E).

### 3.4. In Vivo Regeneration of Muscle Defects by Human Pluripotent Stem Cell-Seeded Scaffolds

To assess the regenerative capacity of PAX7::GFP-sorted hPDMs grown on fibrin scaffolds, they were tested by implantation into subcritical tibialis anterior (TA) defects (n = 9 mice) (Figure 4A) and removed with the TA muscle for histological examination after 1 week. Masson’s trichrome of the treated TA muscle demonstrated moderate fibrosis with apparent growth of muscle into the defect area (Figure 4B). The fibrotic region of the defect accounted for 7.51 ± 3.57% of the defect regions. When cut longitudinally, samples demonstrated limited survival of human cells, as indicated by the white staining of LaminAC+ human nuclei. F-actin staining indicated the native muscle (Figure 4C, lower right) and cell–body structure in the fiber construct. In cross-section, we observed significant numbers of centrally located nuclei and eMHC staining indicated the regenerating region of muscle and vascular ingrowth into the defect region (Figure 4D). Several of the remaining human cells (LamAC) were associated with eMHC+ regions (Figure 4D).

### 3.5. Comparison of Regeneration Outcomes of Skeletal Muscle Defects by Human Progenitor-Seeded Scaffolds

Our group has used other human [22] and non-human cells [3] in a similar manner to regenerate skeletal muscle tissue in VML defects. We accessed tissue sections from these prior studies and stained them for comparison with samples from this current study. Human adipose-derived stem cells (hASCs) have demonstrated the ability to support myogenic differentiation, but tissue regeneration is characterized by substantial scar tissue and amorphous architecture (Figure 5). Compared to hASC-seeded fibers, unsorted hPDM and PAX7::GFP-sorted hPDM-fibers demonstrate more organized regeneration. Mouse C2C12s have shown the greatest tendency for cellular proliferation with minimal fibrosis. With a disparity between human and mouse cells in our mouse model, the response of host macrophages was characterized. F4/80 was used as a pan-macrophage marker, and CD206 was used as an M2 phenotypic marker.

## 4. Discussion

Obtaining myogenic progenitors derived from human pluripotent stem cells in an efficient and reproducible manner is central to effectively engineering 3D human skeletal muscle grafts. Our transgene-free differentiation protocol can induce hPSCs to the myogenic lineage and can produce large amounts of expandable myogenic progenitors [12]. We further corroborated the identity of these cells using a singleCellNet classifier [32], which provided an unbiased categorization of both unsorted and PAX7::GFP-sorted cells as predominantly embryonic myocyte-like based on their transcriptional profiles from bulk RNA sequencing. The bulk RNA sequencing data in combination with in vitro immunostaining indicated that the unsorted hPDMs had a sizeable non-myogenic cell population that may have included undifferentiated embryonic cells, neural crest cells, and FAPs. However, the unsorted hPDMs still classified as fetal myocytes, likely due to the existing PAX7+ population, cells in the PAX7 lineage that had already or have yet to express PAX7, and non-PAX7 expressing cells that may have undergone myogenic differentiation [34]. The fluorescently labeled PAX7 reporter cells enabled the selection of a more homogenous population. Although myogenic progenitors do not necessarily require PAX7 expression [34], it has been shown that embryonic PAX7+ cells give rise to muscle progenitors and most satellite cells [35]. In our cultures, PAX7::GFP-sorted hPDMs classified as ‘more myogenic’, based on gene set enrichment analysis.

Based on our previous findings, we surmised that the fibrin microfiber bundles with topographical alignment and myomimetic stiffness would provide a myoinductive microenvironment to study human skeletal muscle development. The biophysical cues provided by the electrospun fibrin microfiber bundles were sufficient to instruct PAX7::GFP-sorted hPDMs to form aligned, multinucleated myotubes [36,37,38]. The cultures resulted in cells that stained positively for desmin and MHC while retaining a minute PAX7+ population, similar to the native muscle, confirming the suitability of the scaffold for engineering 3D human skeletal muscles. Future studies may focus on providing further maturation of the grafts, which might be achieved through extended culture periods and the application of tensile strain [14] as we did not observe spontaneous contraction of the 3D muscle grafts. As in previous studies with C2C12s [14] and human adipose-derived stromal/stem cells (hASCs) [3], the tissue formed primarily at the periphery of the scaffolds. However, the hPDMs appeared to proliferate less than these other cell populations, which may have affected their in vivo fate. The fibrin microfiber bundles have been explored for muscle regeneration in the mouse TA model of VML. We previously showed that the microfiber bundles enabled functional muscle regeneration with little-to-no fibrosis and robust vascular infiltration when used with murine myoblasts [3] while collagen deposition by hASCs insulated the fibers, resulting in minimal myogenic regeneration [22]. These were not evident with PAX7::GFP-sorted hPDMs-derived muscle grafts, which also had minimal fibrosis around the grafts and high vascular infiltration at the interface with native muscle. However, the regeneration result fell short of what we previously observed with C2C12s.

In our studies, we noted a low cell retention relative to other published studies [4,23,39], despite utilizing a high seeding density, compared to previous work [4,23]. However, we recorded a low seeding efficiency (~5–20%) and our method of seeding cells after fabricating the scaffolds resulted in cells remaining primarily on the surface rather than throughout the scaffold cross-section [14], as occurs when cells are encapsulated directly in the bulk hydrogel. This low cell survival may indicate that a critical mass of cells—which may not have been achieved with our constructs—is required for post-transplantation survival. In ongoing studies, we are exploring methods to improve the density of cells in the interior of the microfiber bundles by including them in the electrospinning process [40]. We, therefore, employed a subcritical defect and assessed at an early (1 week) time-point to maximize our observation of human cells. Although we observed limited cell survival, many of the human cells were associated with eMHC+ regions. We speculated this low survival might be related to decreased heterogeneity of the cells, as native skeletal muscle has several mononuclear cell types that support the survival and maintenance of the myogenic population [41,42]. FAPs have recently been shown to be necessary for muscle stem cell proliferation following injury and prevention of muscle cell atrophy [43]. Despite the low survival of transplanted cells, we observed a high level of cellularity in the defect. The mice lacked mature T and B cells, as well as complement and natural killer cells, while dendritic cells and (functionally impaired) macrophages are present [44,45]. Most of the cells in the defect stained positively for macrophage markers. Though many of the PAX7::GFP-sorted cells seemed to die shortly after implantation, their clearance may still polarize macrophages and induce a therapeutic effect [46]. Future investigations into controlled mixtures of cell populations could be advantageous.

## 5. Conclusions

Combining PAX7::GFP-sorted hPDMs and electrospun fibrin microfiber bundles facilitates the formation of human skeletal muscle constructs with myotube alignment, expression of myogenic markers, and retention of a small PAX7 population. The system is advantageous for in vitro experiments focused on studying 3D-engineered human muscle formation, the interaction of PAX7 progenitors with mature myotubes, and cell–cell signaling interactions among myogenic and non-myogenic phenotypes. In vivo studies of regeneration following skeletal muscle injury strongly indicate a need for increasing cell density on our scaffolds to promote greater survival of transplanted cells. However, these studies also demonstrated the complexity in fine-tuning the engineered grafts to maximize regenerative outcomes: Grafts engineered from PAX7::GFP-sorted cells showed low scarring, high vascular infiltration into the defects, and elicited a pro-regenerative immune response. Future studies to enhance skeletal muscle regeneration with PAX7::GFP-sorted hPDMs will focus on increasing the cell numbers and uniformity throughout the scaffolds, investigating signaling interactions between myogenic and non-myogenic cells to better optimize cell ratios, and modifying the in vivo microenvironment to enhance cellular activity.

## Figures and Tables

**Figure 1 bioengineering-09-00693-f001:**
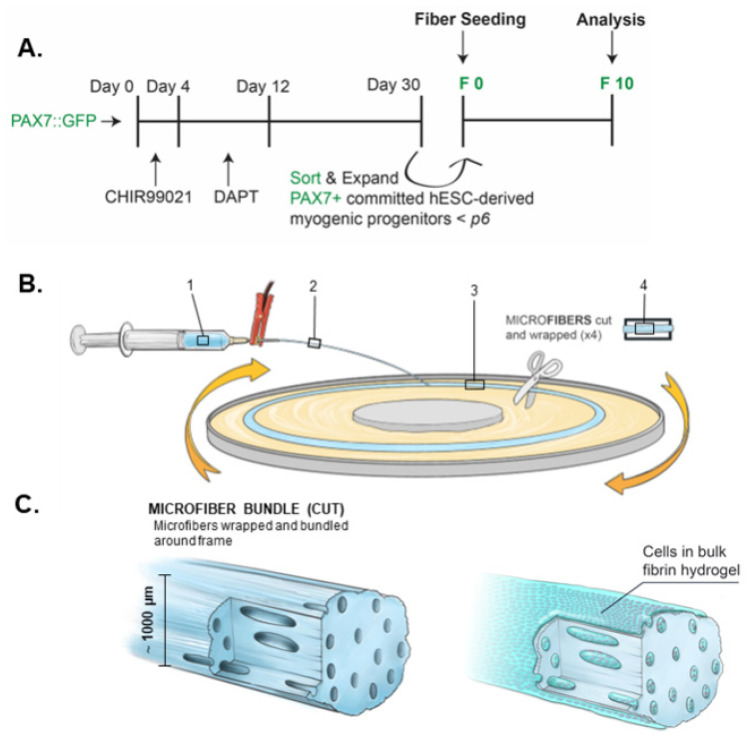
Induction of hPDMs from pluripotent stem cells and seeding onto fibrin microfiber bundles. (**A**) Timeline of myogenic commitment of hPSCs and fiber seeding. (**B**) Hydrogels are loaded into syringes and voltage applied across needle tip (1) resulting in a continuous stream of extruded hydrogel (2). Fiber is formed on a rotating flatform (3), then spun onto a frame (4). (**C**) Resulting fibers are approximately 1000 µm. Cells were also seeded in bulk hydrogel onto microfiber bundles.

**Figure 2 bioengineering-09-00693-f002:**
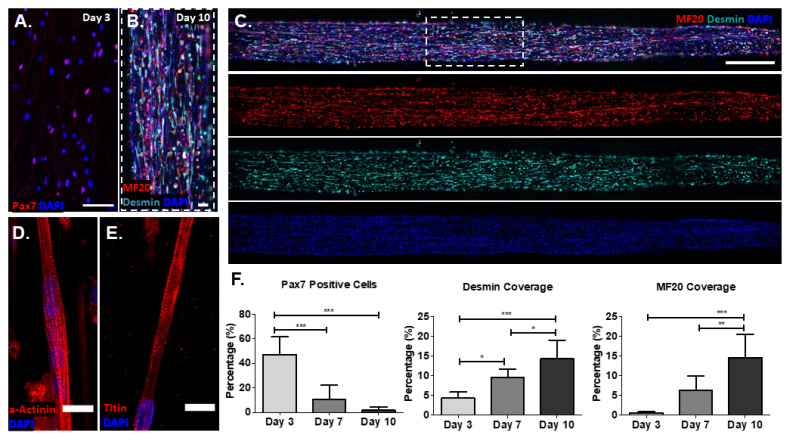
PAX7-sorted myogenic progenitors form engineered 3D skeletal muscle grafts. (**A**) Day 3 expression of Pax7 in sorted myogenic progenitors grown on fibrin microfiber bundles (scale bar: 100 µm). (**B**) By day 10, sorted myogenic progenitors fuse to form desmin and MHC positive myotubes (scale bar: 100 µm). (**C**) Desmin and MHC expression are consistent along the entire length of the fibers (scale bar: 1 mm). (**D**,**E**) Striated myotubes expressing a-actinin (**D**) and titin (**E**) (scale bars: 20 µm). (**F**) Quantification of pax7 desmin, and MHC (MF20) expression over time from myogenic progenitors forming engineered skeletal muscle. *: *p* < 0.05; **: *p* < 0.01; ***: *p* < 0.001.

**Figure 3 bioengineering-09-00693-f003:**
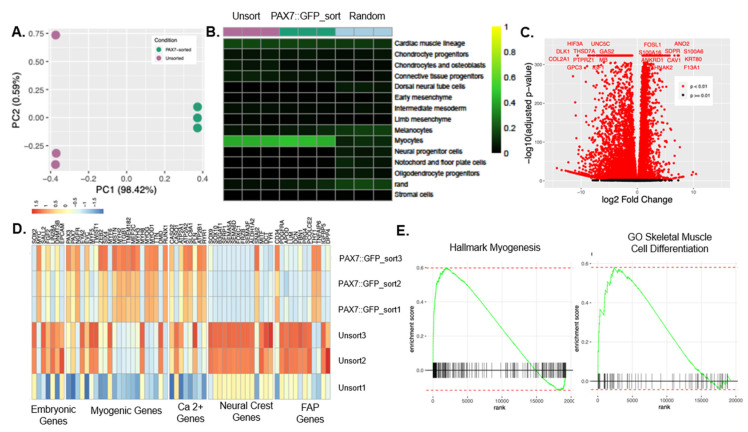
RNA Sequencing elucidates differences between Pax7-sorted and unsorted myogenic progenitors. (**A**) Principal component analysis demonstrates Pax7-sorting selects for a distinct sub-population. (**B**) CellNet/SingleCellNet tool allows for the comparison of sorted and unsorted populations to in vivo cells from mouse organogenesis cell atlas fetal days 9.5–13.5. Pax7-sorted cells show a slightly higher fetal myocyte classification score than unsorted cells. (**C**) Volcano plot shows the distribution of genes more highly expressed in unsorted cells (**left**) and Pax7-sorted cells (**right**). Red dots indicate significantly different expression. Labeled genes are the 12 most significantly differentially expressed. (**D**) Heatmap of key embryonic, myogenic, Ca2+ signaling, and neural crest-related genes expressed in unsorted and sorted populations indicates a higher overall expression of myogenic genes in Pax7-sorted cells and embryonic and neural crest-associated genes in unsorted cells. (**E**) Relevant gene sets enriched in sorted (Hallmark Myogenesis and GO Skeletal Muscle Cell Differentiation).

**Figure 4 bioengineering-09-00693-f004:**
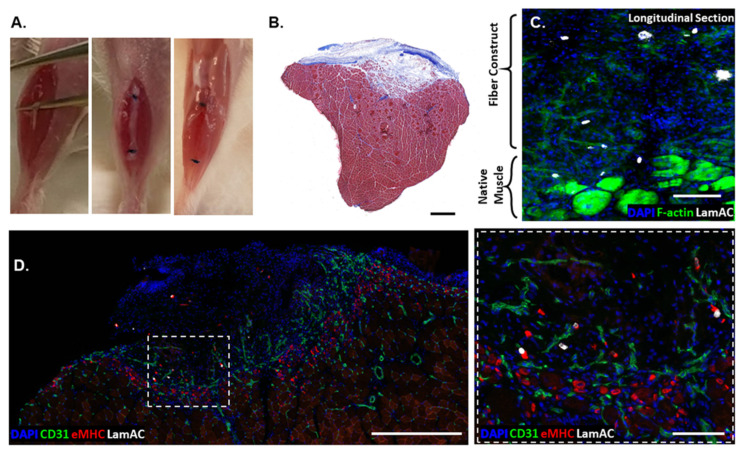
PAX7::GFP-sorted hPDMs were spatially associated with regenerating myofibers in skeletal muscle defects. (**A**) Murine skeletal muscle defect model involving the removal of portion of TA muscle, suturing of two fibers to fill the defect, and removal of the muscle after 1 week in vivo. (**B**) Masson’s trichrome staining demonstrated moderate fibrosis (blue) within and surrounding the defect region. (**C**) Thick (70 μm) longitudinal sections of the injured TA showed human nuclei (white) still associated with the fiber region (Scale bar: 100 μm). (**D**) Human nuclei were also visible in cross-section and were often associated with eMHC+ regenerating myofibers present in a CD31-dense regenerating region. Inset showed zoomed in region in dotted white box (scale bars: 500 μm; high mag image 100 μm).

**Figure 5 bioengineering-09-00693-f005:**
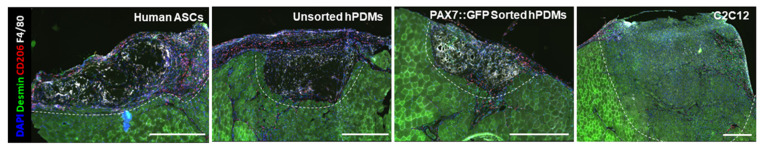
PAX7::GFP-Sorted hPDMs demonstrate moderate regeneration without distortion of native architecture. Human unsorted hPDMs demonstrate moderate regeneration of defect area but showed less organization and increased distortion and growth over native tissue relative to PAX7::GFP-sorted hPDMs. Regardless of human cell source, regeneration is characterized by a robust macrophage response. (scale bars: 500 µm).

## Data Availability

The raw/processed data required to reproduce these findings cannot be shared at this time due to technical or time limitations.

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
