# Peer review of "Engineering Skeletal Muscle Grafts with PAX7::GFP-Sorted Human Pluripotent Stem Cell-Derived Myogenic Progenitors on Fibrin Microfiber Bundles for Tissue Regeneration"

_bioengineering, 2022, doi:10.3390/bioengineering9110693_

Round 1

Reviewer 1 Report

This is a traditional study of this research group. It is well planned and executed to a high standard. However, I have a few clarifying questions and comments:

1. On fig. 2, 4 and 5 the DAPI inscription is practically invisible, it needs to be fixed.

2. Fig. 4. Is it possible to quantify fibrosis data?

3. In addition, did the authors evaluate the level of centrally nucleated fibers, which reflects the efficiency of muscle tissue regeneration. Is this approach applicable in this case?

4. I didn't find any mention of how many animals were used in the experiments?

5. What is volumetric muscle loss (VML)? It needs to be explained to the reader. What can be its cause and what are the existing methods of treatment?

Author Response

The authors would like to thank the reviewers for their critical review of the manuscript and for their helpful comments. We have addressed each comment and have revised the manuscript based on these suggestions. We believe these changes strengthen the manuscript. The changes are highlighted in red font in the revised manuscript.

Reviewer 2 Report

This is very interesting paper regarding that the biophysical cues provided by the microfiber bundles can induce unsorted hPDMs to form multinucleated myotubes that express desmin and myosin heavy chain.  The system looked to be advantageous for in vitro experiments focused on studying 3D engineered human muscle formation, the interaction of PAX7 progenitors with mature myotubes, and cell-cell signaling interactions among myogenic and non-myogenic phenotypes.  I have few concerns as below.

Summary figure should be included in Conclusion section.

Please carefully check reference format (eg., abbreviated journal names, no italic for paper title, etc.)

A lot of typo errors should be revised (eg., in vivo --> to be italic, L384, space should be removed)

Authors should show levels of molecular parameters on muscle regeneration and differentiation biomarkers.

Authors should show whether there are infiltration of immune cells by showing immunohistochemistry.

Author Response

(The authors gave the same response as above.)

Reviewer 3 Report

The topic of this paper is quite interesting. I suggest its publication after following issues are well addressed.

1. In introduction, please highlight the importance of the study of this paper and the differences between this work and other literature for more comparison and discussion.

2. The introduction should give a brief description of the key research work of this paper and highlight the strengths of this work.

3. In section 2.2 of “ hPSC Myogenic Commitment and Expansion”, it should be fully described the experimental process by which human embryonic stem cells (H9-hESCs; WiCell) are cultured and transformed to the adult muscle lineage.

4. In section 2.9 of “RNA Sequencing Analysis”, the basis for the FGSEA parameters to do GSEA runs should be mentioned in the experimental section.

5. “Expression of PAX7 decreased significantly over the 10-day growth period while desmin and MHC coverage significantly increased over the growth period (Fig. 2F).” It is only the experimental results that are described here, without a full explanation of this result that was generated.

6. The dRNA sequencing in Figure 3 only elucidates the differences between pax7 classified and unclassified myogenic progenitor cells, but it does not fully explain the reasons for the differences.

7. Abstract and conclusion should be improved by highlighting the aim of the paper and summarizing the used methods and the obtained results.

Author Response

(The authors gave the same response as above.)

Round 2

Reviewer 1 Report

The authors adequately responded to my comments and improved the manuscript.

Reviewer 2 Report

This paper is now acceptable.

Reviewer 3 Report

Can be accepted now.